# The Effect of Weaning with Adult Food Typical of the Mediterranean Diet on Taste Development and Eating Habits of Children: A Randomized Trial

**DOI:** 10.3390/nu14122486

**Published:** 2022-06-15

**Authors:** Raffaella de Franchis, Luigi Bozza, Pasquale Canale, Maria Chiacchio, Paolo Cortese, Antonio D’Avino, Maria De Giovanni, Mirella Dello Iacovo, Antonietta D’Onofrio, Aniello Federico, Nicoletta Gasparini, Felicia Iaccarino, Giuseppe Romano, Raffaella Spadaro, Mariangela Tedesco, Giuseppe Vitiello, Angelo Antignani, Salvatore Auricchio, Vincenzo Valentino, Francesca De Filippis, Danilo Ercolini, Dario Bruzzese

**Affiliations:** 1Italian Federation of Maedical Paediatrics (FIMP), 80142 Naples, Italy; luigibozza1963@virgilio.it (L.B.); pasqualecanale03@gmail.com (P.C.); mariachiacchio@tiscali.it (M.C.); pacortese@libero.it (P.C.); davinoant@gmail.com (A.D.); mdegiovanni@hotmail.it (M.D.G.); delloiacovom@gmail.com (M.D.I.); antonietta.donofrio55@gmail.com (A.D.); federicoaniello116@gmail.com (A.F.); nicolegasparini@libero.it (N.G.); iaccarino.felicia@gmail.com (F.I.); pino2ro@yahoo.it (G.R.); raffyspadaro@gmail.com (R.S.); tedescomariangela@alice.it (M.T.); dottvitiellogiuseppe@tin.it (G.V.); 2Department of Clinical Medicine and Surgery, University of Naples Federico II, 80131 Naples, Italy; angelo.antig@gmail.com; 3European Laboratory for the Investigation of Food-Induced Diseases, Department of Medical Translational Sciences, University of Naples Federico II, 80131 Naples, Italy; salauric@unina.it; 4Department of Agricultural Sciences, University of Naples Federico II, 80055 Portici, Italy; vincenzo.valentino2@unina.it (V.V.); francesca.defilippis@unina.it (F.D.F.); danilo.ercolini@unina.it (D.E.); 5Task Force on Microbiome Studies, University of Naples Federico II, 80100 Naples, Italy; 6Department of Public Health, University of Naples Federico II, 80131 Naples, Italy; dario.bruzzese@unina.it

**Keywords:** complementary feeding, Mediterranean Diet, gut microbiota, fresh foods, taste development

## Abstract

Mediterranean Diet (Med Diet) is one of the healthiest dietary patterns. We aimed to verify the effects of weaning (i.e., the introduction of solid foods in infants previously fed only with milk) using adult foods typical of Med Diet on children eating habits, and on the microbiota composition. A randomized controlled clinical trial on 394 healthy infants randomized in a 1:1 ratio in a Med Diet group weaned with fresh; seasonal and tasty foods of Med Diet and control group predominantly weaned with industrial baby foods. The primary end point was the percentage of children showing a good adherence to Med Diet at 36 months. Secondary end points were mother’s changes in adherence to Med Diet and differences in children gut microbiota. At 36 months, children showing a good adherence to Med Diet were 59.3% in the Med Diet group and 34.3% in the control group (*p* < 0.001). An increase in adherence to the Med Diet was observed in the mothers of the Med Diet group children (*p* < 0.001). At 4 years of age children in the Med Diet group had a higher gut microbial diversity and a higher abundance of beneficial taxa. A Mediterranean weaning with adult food may become a strategy for early nutritional education, to develop a healthy microbiota, to prevent inflammatory chronic diseases and to ameliorate eating habits in children and their families.

## 1. Introduction

The association between systemic chronic inflammation and chronic diseases such as obesity, diabetes, cancer, autoimmune diseases, inflammatory bowel diseases, several ageing linked diseases, etc., is now widely accepted [1,2]. Due to the economical impact of all these diseases in all Health Systems, a wide strategy to prevent the risk of developing them is strongly needed [2]. Among the most common triggers of systemic chronic inflammation, diet and intestinal dysbiosis seem to play a crucial role [2]. There is also increasing evidence that diet components are directly involved in the pathogenesis of chronic inflammatory diseases by shaping the composition of the microbiota [3,4]. Med Diet not only has been allocated among anti-inflammatory nutritional programs and has been proposed as a food-based approach to prevent and treat chronic inflammatory diseases but also has been related to beneficial effects on the gut microbiota [5].

Another aspect to be considered when focusing on the role of the Med Diet in preventing chronic diseases is the timing of introduction of this style of eating. In fact, the earlier the gut exposition to the Med Diet foods the better the development of a safe microbiota [6]. We are also aware of the impact of pre and perinatal exposition to some tastes on the later food acceptance [7,8]. 

In support of these assertions, some evidence shows that complementary responsive feeding practices, such as baby-led weaning (associated with early satiety-responsiveness acquisition), are protective against obesity respect to usual complementary feeding mode [9]. In contrast, no data are available on the effects of an early introduction of Med Diet in children but a possible protective role against celiac disease [10] and inflammatory bowel diseases [11,12] has been demonstrated after an early introduction of a dietary pattern very similar to Med Diet.

Due to all the evidence of the positive effects of eating according to a Med Diet style, we would expect a high diffusion of it among the entire population. In contrast to what expected, several studies have showed a very poor adherence to it in children or adolescents [13,14,15,16] and to our knowledge, no intervention study has been carried out on a population of children between 0 and 3 years of age to verify the medium-term effect of early nutrition in terms of food habits, BMI and microbiota composition.

The aim of our work was to verify if weaning (i.e., the introduction of solid foods in infants previously fed only with milk, currently known as complementary feeding) with natural foods of the Med Diet [17] is a simple and easily reproducible approach to increase in children (and their families) the adherence to it and to positively affect the gut microbiota.

## 2. Materials and Methods

This study was registered at ClinicalTrials.gov (NCT05297357).

18 family pediatricians participated in the study. Each pediatrician enrolled 10/15 infants between 4 and 6 months of age. Infants, either breast fed or formula fed between 4 and 6 months of age were included in the study. Informed consent from at least one parent was mandatory. Infants either with associated comorbidities or with low birth weight were excluded. All pediatricians preliminarily defined their methodology of measuring kids’ weight and length according to standard procedures [18] and each child was always measured by the same pediatrician. They also standardized their Mediterranean weaning schemas. Anthropometric measures (weight, length and head circumference) of each infant were taken during visits at enrollment (T0), and at 12 (T12), 24 (T24) and 36 (T36) months of age. Parents’ socio-demographic characteristics were registered at the beginning of the study.

The adherence to Med Diet of both child and his/her mother was investigated by using two age-related questionnaires: the KidMed questionnaire [19] at 12, 24 and 36 months of age and the adult questionnaire [20] at 12 and 36 months of age of their kid.

### 2.1. Interventions

Each infant was randomly assigned to one of the following arms:

Control arm: kids were weaned using a “traditional” schema, that had been used for years in the pediatricians’ clinical practice, mainly based on industrial baby foods, with very poor content of fresh foods, late use of legumes (7–8 months) and late introduction of fresh fish (after one year). Salt and cooked oil were added after one year of age. 

Experimental arm: kids were weaned with a “Med Diet” schema (MD Foundation-2011) [17] characterized by using only fresh foods traditionally used in Med Diet and appropriately adapted to infants (smoothie or crashed foods). Seasonal fruit and vegetables, including broccoli and cauliflower were proposed as a puree since the early phases of weaning. Various kinds of fresh blue fish (including anchovies, mackerel, flag fish) as well as cod and sole, all flavored with garlic and cherry tomatoes, were proposed at 7 months of age. Spices and herbs such as thyme, marjoram, rosemary, parsley, as well as garlic and onion were regularly used to flavor the meal. No salt was used while 2 g of Parmesan cheese was added to make the meal tangy. No sugar was added to any meal. Due to the low age of children, some differences were applied in respect of the original Med Diet schema: nuts only after one year of age; total exclusion of red wine, only one serving of vegetables, fruit and cereal per meal and reduced number of eggs to 1–2 weekly. Finally, our weaning schema did not include any sweet. 

All kids, either belonging to the experimental or to the control arm were authorized to enter the family diet and use salt and cooked oil, at 1 year of age. At this time, families of babies in the experimental group were much more aware of the correct Mediterranean eating style.

Both weaning schemas followed current guidelines as far as lipid, protein, carbohydrates and caloric content recommended for that age [21]. The constant supervision of a nutritionist allowed to monitor the adherence of all children to guidelines, either when eating a traditional weaning schema or when following the Mediterranean one.

Despite the use of fresh foods only in the Med Diet weaning, another major difference between the two eating schemas, either traditional or Mediterranean, was the way of cooking foods. Mothers of the treatment arm were encouraged to cook, starting from the first meal at weaning, in a tasty way and to taste the meal before offering it to the kid. Although salt was not used, spices and herbs adjusted the flavor toward dishes usually appreciated by adults. Only the experimental group received a reinforcing message on nutritional education toward the Med Diet at each access to the pediatrician’s office, even occurring for other clinical problems. The reinforce typically last five minutes and had the purpose to verify if correct eating habits were still kept focusing on the consume of fresh fruits and vegetables even with green leaves, fish and avoiding any kind of sweet.

A subset of randomly chosen children in Med Diet group (*n* = 26) and in the control group (*n* = 25) were studied in their gut microbiota composition at 4 years of age (mean ± standard deviation: 4.2 ± 0.6 years—min: 3.3; max: 5.6 years).

### 2.2. Inclusion and Exclusion Criteria

Inclusion criteria were as follows: healthy infants, stable clinical conditions and feeding by mouth with human milk or formula. Exclusion criteria were the presence of any comorbidity including prematurity.

### 2.3. Ethical Approval and Informed Consent

Written informed consent was obtained from all the mothers of the enrolled children before the study. Procedures followed were in accordance with the ethical standards of the responsible institutional committee on human experimentation (“Comitato Etico Federico II”-213/20).

### 2.4. Study Objectives and Endpoints

The primary aim of this RCT was to assess the effects of a weaning scheme based on fresh, seasonal and natural foods of the Med Diet on the medium-term adherence of the children to the Med Diet.

Secondary aims were to assess the effect of such weaning scheme on: the BMI of children at 36 months of age; the eating habits of children over the time; mother’s changes in adherence to Med Diet by comparing their habits registered at 12 months and at 36 months of age of their kids; differences in gut microbiota composition in a subset of children at 4 years of age.

All primary and secondary endpoints refer to the third assessment of follow-up which was planned once children reached the 3 years of age (T36). Due to organizational difficulties encountered during COVID-19 pandemics, the fecal sample collection was delayed and took place approximatively one year late.

The primary efficacy end point was the percentage of children showing a good adherence to the Med Diet, defined as a score in the KidMed questionnaire ≥8 [19]. 

Secondary end points were the mean of the KidMed score, the average children’s BMI, the percentage of overweight and obese children, the average mother’s adherence to Med Diet and the gut microbiota composition. 

Overweight and obesity were defined according to extra-centiles of the Italian BMI growth norms published in Cacciari 2006 [22]. Mother’s adherence was measured by computing the percentage of “positive” answers to the adult questionnaire, i.e., answers showing a dietary habit consistent with Med Diet, with respect to the total number of provided answers.

### 2.5. Sample Size

Assuming a precautionary percentage of good adherence in the control group equal to 50% and considering as clinically relevant a difference between groups equal to 17%, which correspond to two-fold odds of good adherence in the experimental group, a sample size of 175 children per treatment arm was deemed sufficient to assess such difference, if truly exists, with a two sided alpha equal to 0.05 and 90% of power. Considering a potential dropout rate equal to 15%, approximately 200 children per treatment arm should be enrolled.

### 2.6. Randomization

Randomization was centralized according to a block scheme (blocks of 6 and 4 children which were randomly alternated). Randomization lists were generated using the sample(...) function of R language. Each family pediatrician received a concealed list of at least forty codes and, with respect to the pediatricians who agreed to participate to the study, enrollment was competitive until the achievement of the planned sample size.

### 2.7. DNA Extraction and Fecal Microbiota Analysis

At 4 years of age, fecal samples were collected from 25 cases and 26 controls to evaluate differences in the gut microbiota composition.

Fecal samples were collected at home, following the Standard Operating Procedure 04 (SOP04) by the International Human Microbiome Standard (IHMS) Consortium. Briefly, samples have been kept at 4 °C and transported to the laboratory within 24 h, where they have been stored at −80 °C prior to further analyses. Microbial DNA extraction was performed following IHMS SOP07_V2, starting from 200 mg of frozen fecal sample.

Bacterial diversity was assessed by high-throughput sequencing of the amplified V3-V4 regions of the 16 S rRNA gene (about 460 bp). PCR was carried out with the primers S-D-Bact-0341-b-S-17/S-D-Bact-0785-a-A-21, as previously described [23]. PCR products were purified with Agencourt AMPure beads (Beckman Coulter, Milan, Italy) and quantified using an AF2200 Plate Reader (Eppendorf, Milan, Italy). Equimolar pools were sequenced on an Illumina MiSeq platform, yelding to 2 × 250 bp, paired-end reads.

### 2.8. Statistical Analysis

All statistical analyses were conducted using the platform R (vers. 4.0; The R Foundation for Statistical Computing).

The primary analysis population was the Modified Intention to Treat (ITT) population, defined as all randomized children who received the assigned weaning scheme from their family pediatrician and for whom outcome data were available. For the assessment of the primary endpoint a sensitivity analysis was conducted in which missing outcome data were imputed using a multiple imputation approach. Variables entered in the imputation model were, besides treatment arm, baseline mother’s adherence to Med Diet, mother’s degree, children’s weight at enrollment, family structure and feeding at enrollment.

Demographic and clinical baseline data were summarized using standard descriptive statistics and compared between group (without reporting statistical significance) to assess whether good balance was achieved by randomization.

The assessment of the primary outcome was based on a chi square test and treatment effect was quantified computing the Odds Ratio (OR) between the good adherence and treatment arm with the corresponding 95% Confidence Interval (95% CI). The same approach was adopted for all secondary endpoints expressed on nominal scale. Numerical scores of children’s and mother’s adherence to Med Diet as well as children’s BMI were analyzed using a T test for independent samples. Longitudinal trajectories of children’s adherence were analyzed using Linear Mixed Models (LMM) using time as categorical factor to allow for non-linear effects; results of LMM were expressed as Estimated Marginal Means (EMMs) with the corresponding 95% CIs.

### 2.9. Fecal Microbiota Analysis

Sequences were sent to QIIME2 q2cli v2020.04 and the plugin DADA2 was used to trim primers and low-quality bases, denoise and merge forward and reverse reads. Chimeric sequences were found and filtered through the options “—*p*-chimera-method pooled” and “—*p*-min-fold-parent-over-abundance 10”.

Taxonomy was inferred by mapping the representative sequences against the Greengenes database (release 13_8) through the ‘consensus_vsearch’ method included in the ‘feature-classifier’ plugin. The resulting ASV (Amplicon Sequence Variants) table was imported into a R environment (version 3.6.3) for statistical analysis and visualization. Samples were rarefied at the same number of sequences, then alpha-diversity indices were estimate using the phyloseq R package (function plot_richness).

Boxplots were drawn by using the function geom.boxplot from the ggplot2 R package, while the heatmap was obtained through the function pheatmap (pheatmap R package). Differences in the overall microbiota composition between the two groups were evaluated by Permutational multivariate analysis of variance (nonparametric MANOVA) based on Bray-Curtis distance matrix carried out in R environment (adonis function in vegan package). Pairwise Wilcoxon tests were used to determine significant differences in alpha diversity parameters or in the abundance of specific taxa.

## 3. Results

### 3.1. Baseline Characteristics

From May 2015 to July 2016, 394 children were enrolled; 194 (49.2%) were assigned to the experimental arm and 200 (50.8%) to the control group (Figure 1). 

Mean age at enrollment was 145.4 ± 19.2 days (from 47 to 197 days). Breastfeeding was present in 112 children (28.5%) and 73.3% of the mothers received, at most, a secondary school certificate. The Med Diet adherence of the mothers of the enrolled children, as measured by the adult questionnaire, was equal to 62.5 ± 13.7%. Clinical and demographical characteristics of both children and their parents, stratified by treatment arm, are shown in Table 1. No clinically relevant differences were observed between the two groups at baseline.

### 3.2. Mediterranean Diet Adherence in Children

At the third follow-up visit, 358 (90.9%) children had available data for the primary endpoint analysis. Mean age of children at the end of follow-up was equal to 36.9 ± 3.2 months (min: 23.2, max: 52.2). 

The percentage of children with a good adherence to Med Diet was equal to 59.3% in the experimental group and 34.3% in the control group (OR: 2.80; 95% CI: 1.82 to 4.30; *p* < 0.001). KidMed score was significantly higher in the Med Diet group with respect to the control group (7.5 ± 3 vs. 6.0 ± 2.9; 95% CI: 0.9 to 2.1; *p* < 0.001) (Table 2).

With respect to the single items of the KidMed questionnaire, children in the experimental group showed higher regular consumption of fresh and cooked vegetables (90.8% vs. 74.5%; *p* < 0.001), fish (79.8% vs. 64.2%; *p* = 0.002) and nuts (41.1% vs. 25.5%; *p* = 0.003) while were less likely to eat fast-food (9.8% vs. 20%; *p* = 0.013), baked good and pastries for breakfast (55.2% vs. 74.5%; *p* < 0.001) and sweets (35.6% vs. 57.6%; *p* < 0.001) (Figure 2).

Longitudinal trajectories of Med Diet adherence showed that difference between groups was not statistically significant in the first assessment, corresponding to an average age of 12.5 ± 1.8 months, but became significant from the second assessment corresponding to an average age of 24.8 ± 2.9 months (Figure 3).

### 3.3. Children’s BMI

No significant differences between groups were observed with respect to BMI (16.4 ± 2 in control group and 16.2 ± 1.5 in the experimental group; *p* = 0.413). Percentages of overweight and obese children were lower in the Med Diet group, but this difference did not reach statistical significance (Table 2).

### 3.4. Mediterranean Diet Adherence in Mothers

Mothers’ adherence to Med Diet was evaluated at T12 and at T36 of their kids. Differences in eating behavior were analyzed using the adult questionnaire for Med Diet adherence.

In the second assessment, mothers of children in the Med Diet group showed a significant higher adherence to the Med Diet as measured by the adult questionnaire when compared with mothers of children of the control group (70.3% ± 18.2% vs. 61.7% ± 17.1%; 95% CI: 4.9% to 12.3%; *p* < 0.001) (Table 2) with an higher consumption of olive oil, vegetables, fish, nuts and white meat and a lower intake of butter and commercial sweet products. Mothers in the control group showed no increase in their adherence (−0.43 ± 19.4; 95% CI: −3.16 to 2.53; *p* = 0.827) while mothers in the experimental group experienced a 7.6% increase in their adherence (95% CI: 4.57% to 10.63%; *p* < 0.001) at 36 months of age of their kids.

### 3.5. Influence of Weaning Type on Gut Microbiota Composition

The gut microbiota composition was evaluated at approximately 4 years of age in a subset of the study population (51 children: 26 in Med Diet and 25 controls). This subset was selected from the whole sample using a simple randomization and its size was based on feasibility. The way of feeding children from the third assessment (approximatively at 36 months of age) to the time of feces analysis was based on the same recommendations used in the first three years of follow-up. The main characteristics of this subgroup are reported in Table 3.

Gut microbiota composition allowed the clustering of all the subjects into two groups (Figure 4).

The percentage of children weaned with Med Diet was 32% in cluster A and 66% in cluster B. Med Diet and control groups showed a significant difference in the overall composition of the gut microbiota at genus level, as defined by Permutational multivariate analysis of variance (nonparametric MANOVA) based on a Bray-Curtis distance matrix (*p* = 0.008, R^2^ = 0.05072, F = 2.6182). Among differentially abundant taxa, *Ruminococcus gnavus* was higher in controls compared to Med Diet, while *Lachnobacterium* and taxa within the *Coriobacteriaceae* family (*Adlercreutzia*, unidentified *Coriobacteriaceae*) were increased in Med Diet (Figure 5A; *p* < 0.05, pairwise Wilcoxon test). In addition, children in Med Diet also showed a significantly higher microbial diversity at 4 years compared with controls (Figure 5B; *p* < 0.05, pairwise Wilcoxon test).

## 4. Discussion

In our study we compared the long- lasting effect on eating habits between children exposed from the weaning to fresh foods of the Med Diet cooked in a tasty way and children weaned with traditional schemas, based on baby foods. Our purpose was to verify the possibility of influencing taste development toward Med Diet foods later in life. We also wanted to investigate the possible role of different weaning schemas in the development of the gut microbiota.

Children who were early exposed to fresh foods of the Med Diet, at 36 months of age, still ate a significant higher amount of fruit and fresh or cooked vegetables per day, when compared to children of the control group. Increase in fish consuming was also observed in children of the experimental group as well as a lower eating of fast foods and baked goods breakfast; they also consumed fewer sweet sugars and sweets during the day.

Comparing the gut microbiota composition, we found that at 4 years of age children weaned with Med Diet showed a difference in the overall composition of the gut microbiota at the genus level, with enrichment of beneficial taxa, and a significantly higher microbial diversity. Despite we are aware of two possible biases, similar to a non-standardized diet in the control group and a constant reinforcement on the educational message in the cases group, all together our strategy led to important changes in the way of eating of children with a long-lasting effect. This effect may have as consequence the reduction of consuming foods typical of the Western Diet thus determining a possible reduction of chronic inflammatory diseases, strictly related to Western Diet. Some evidence, in fact, show that the risk of developing chronic inflammation can be traced back in early infancy giving its effect later in the adulthood health and affecting the mortality risk [2]. An intergenerational transmission risk for SCI has also been hypothesized pointing to a maternal inflammation during pregnancy determining epigenetic modifications to the offspring who will show elevated risk for SCI during childhood and later in adulthood and becoming more exposed to inflammation-related health problems [2].

Children are born with a biological predisposition to prefer sweet [7] and to avoid bitter foods such as green leafy vegetables. This taste predisposition places them at risk for excessive weight gain. Accumulating evidence suggests that, starting before birth and continuing throughout development, there are repeated and varied opportunities for children to learn to enjoy the flavors of healthful foods. Amniotic fluid and breast milk, in turn eases the transition to beneficial foods and this process can continue throughout weaning. These early-life sensory experiences establish food preferences and dietary patterns that set the stage for lifelong dietary habits. The development of evidence-based strategies and programs to facilitate children’s early acceptance of fruit and vegetables are needed [24].

It is well documented the role of gut microbiota dysbiosis in the pathogenesis of chronic inflammatory diseases [25]. In our study gut microbiota composition was evaluated at 4 years in a subset of the studied population and two different patterns in the microbiota composition were identified. Children in Med Diet showed a higher gut microbial diversity and higher abundance of beneficial taxa (e.g., *Coriobacteriaceae*, *Lachnoclostridium*), while *Ruminococcus gnavus* was depleted compared with the control group. Indeed, *Coriobacteriaceae* taxa are able to metabolize polyphenols, rich in the Mediterranean dietary pattern, and produce urolithins, known for the anticancerogenic and anti-inflammatory effect [26,27,28,29]. The abundance of *Coriobacteriaceae* was reported as increased and correlated with urolithins levels upon an intervention with a Med Diet in obese adult subjects, while *R. gnavus* was depleted [30]. *R. gnavus* is usually associated with a diet rich in animal-based products [31] and to inflammatory conditions, such as Chron’s, Inflammatory Bowel Diseases and allergy [32,33,34]. 

Our study has several limitations. First, the gut microbiota was not analyzed at baseline and, due to the organizational difficulty for COVID-19 pandemic, approximatively one year passed between the last follow-up assessment and the feces specimen collection. In addition, the analysis did not involve the entire sample but a relatively small subset of children whose size was determined on a convenience basis. However, as no differences were observed between 3 and 4 years of age in the eating habits of children and the characteristics of the subset resemble those of the entire sample, we are confident that the observed differences in the gut microbiota might be associated with the type of weaning. Another limitation is the decision to perform a randomization at children level and not at pediatrician level by adopting a cluster design that would have reduced the risk of contamination.

Among the factors influencing the gut microbiota composition and related activities, diet plays a primary role. Indeed, the westernization of diet and lifestyle had a tremendous impact on gut microbiota, leading to the loss of microbial diversity [35]. The consumption of a habitual diet rich in vegetable products typical of the Mediterranean area has been demonstrated to boost the development of beneficial microbes in the gut microbiota and the production of health-promoting microbial metabolites [28,31,36]. Interestingly gut microbiota may be implicated in the positive effects of the Med Diet [37,38].

The first years of life are extremely important in shaping the gut microbiota of the infant [39]. In particular, the introduction of solid foods are critical points for the gut microbiota development [40] and the interplay between novel foods introduction, microbial metabolites and immune system may affect the correct development of the immune system and prevent the susceptibility to inflammatory disease in adulthood [41]. Therefore, the design of specific baby weaning schemas can be considered as a tool for the modulation of the gut microbiota and the prevention of immune-related disorders.

In our study, we did not find any difference between cases and controls groups with respect to BMI at 36 months of age. The overall incidence of obesity in the whole sample was much lower than the incidence reported in Italian literature in older children [42]. This finding may probably be related to the too young age of cases and controls at evaluation. The effects on obesity of early eating habits should be evaluated later in life [43]. 

Our results also showed an important effect in changing mothers eating habits in the Med Diet group. When compared to control mothers, cases’ mothers after three years ate much more fruit, vegetables and fish, consumed more olive oil and ate much more dried fruit. On the other hand, they ate much less saturated fats deriving from butter, less industrial pastries. This finding must be taken into account in considering the family pediatrician role in the preventing care system of the entire family, not only of the child. In Italy, the figure of “family pediatrician” in the National Health System may be the key in a “prevention movement” by acting very early on children and following up patients later in life.

We suppose that understanding the remote efficacy, in terms of improving health, of an educational intervention for nutrition, and the possibility of extending that same intervention to populations at risk of diseases influenced by diet, could become, for the family pediatrician, an important field of activity. This could improve pediatric care and enhance the daily clinical research carried out by the pediatricians in their clinical activity, moving towards the “learning health program”, proposed by the NIH Health care system research collaborator [44,45]. In this ambitious path the constant interrelation between family pediatrician, hospital and university, each for their own skills, becomes fundamental.

In conclusion, our work shows that weaning with adult food typical of the Med Diet determines a better adherence to such a diet in the first years of life and enhances a gut microbiota with a higher microbial diversity and a higher abundance of beneficial taxa.

## Figures and Tables

**Figure 1 nutrients-14-02486-f001:**
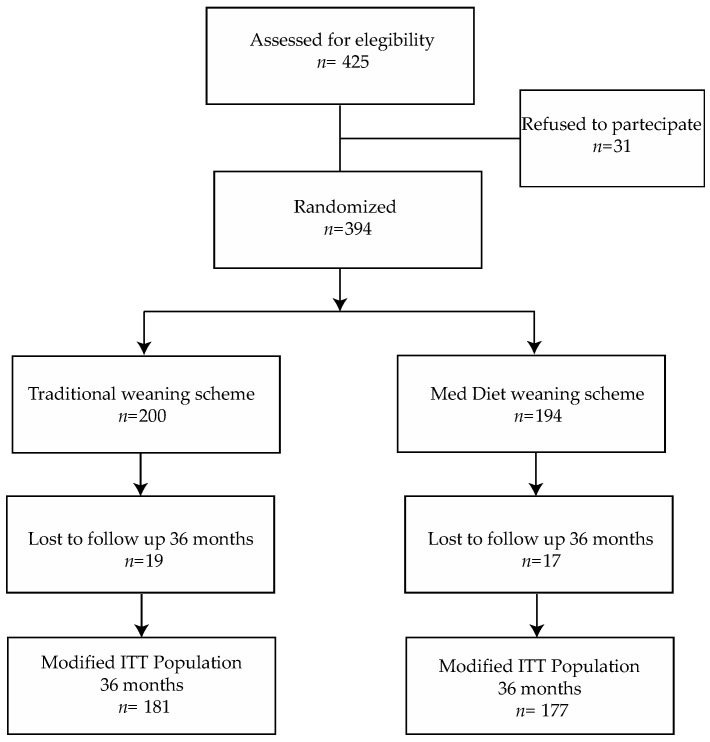
Study Flow-chart.

**Figure 2 nutrients-14-02486-f002:**
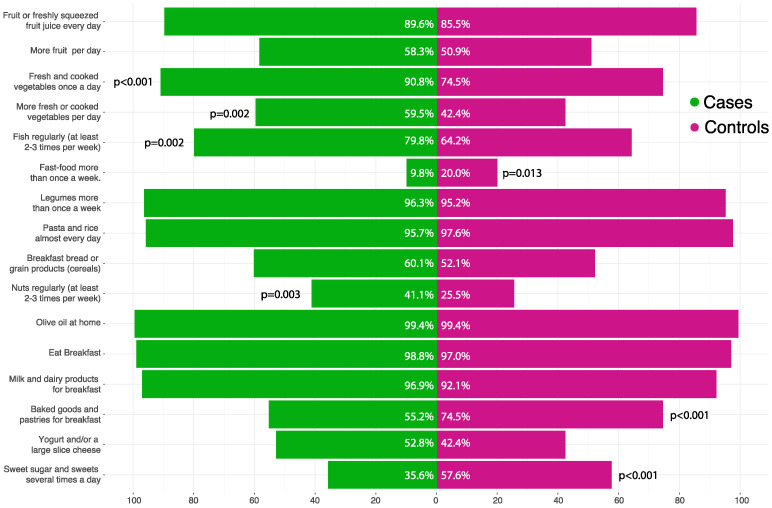
Differences between groups according to the single items of the KidMed questionnaire. Between groups differences for each item were assessed using the Chi square test without adjustment for multiple comparisons.

**Figure 3 nutrients-14-02486-f003:**
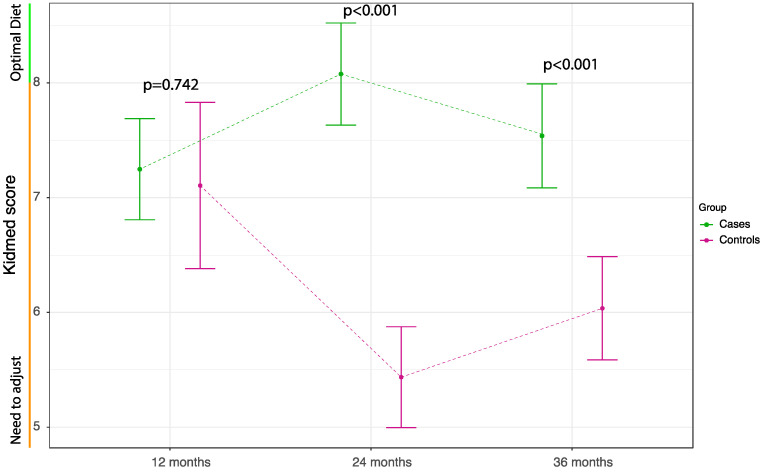
Longitudinal trajectories of KidMed score in children stratified by treatment arm. Results are expressed as Estimated Marginal Means (EMM) with the corresponding 95% Confidence Intervals (95% CIs). EMM’s were based on a linear mixed models using time as categorical factor.

**Figure 4 nutrients-14-02486-f004:**
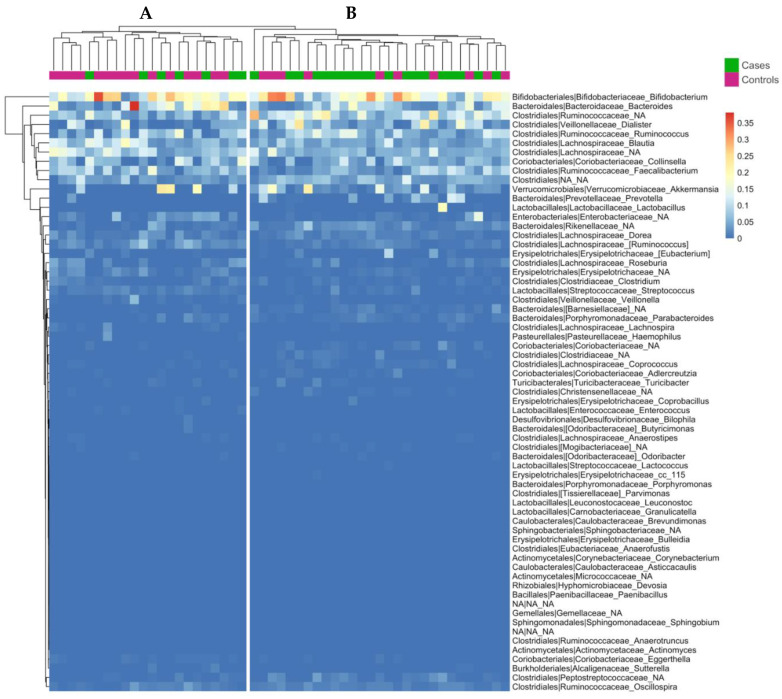
Complete linkage clustering of subjects based on gut microbiota composition at genus level. Only genera with a median absolute abundance >2% were considered. The Canberra metric was used to compute distance between each pair of subjects.

**Figure 5 nutrients-14-02486-f005:**
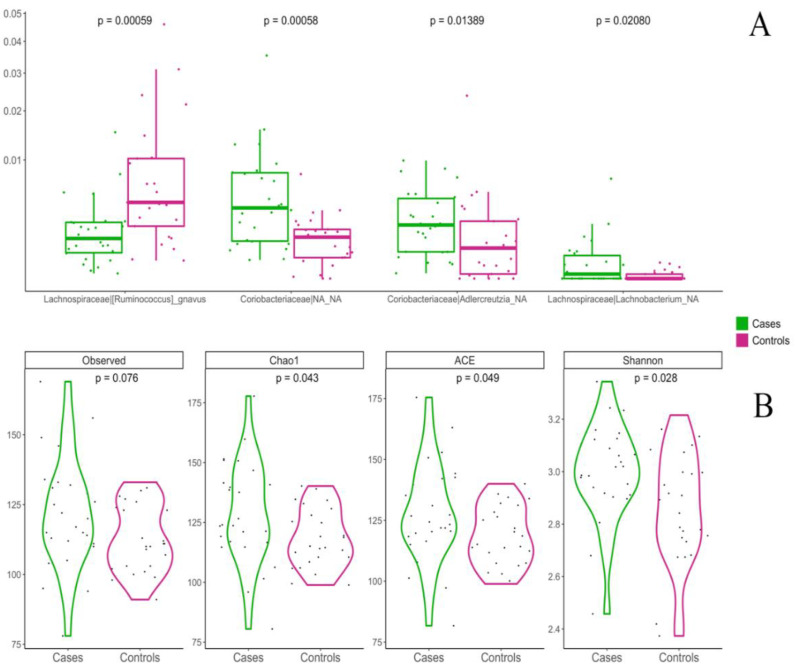
Boxplots showing the relative abundance of microbial taxa between the Med Diet group (green) and controls (pink). (**A**): boxes represent the interquartile range (IQR) between the first and third quartiles and the line inside represents the median (2nd quartile). Whiskers denote the lowest and the highest values within 1.5 × IQR from the first and third quartiles, respectively. Between groups differences were assessed by applying pairwise Wilcoxon test. (**B**): Violin plots showing the difference in four alpha-diversity indices between the Med Diet group and controls. Wilcoxon’s rank sum test was used to assess the significance.

**Table 1 nutrients-14-02486-t001:** Clinical ad demographical characteristics of the sample stratified by treatment arm. Variables are described by mean ± standard deviation (min to max) or absolute frequencies (percentage).

	Control Arm (*n* = 200; 50.8%)	Med Diet Arm (*n* = 194; 49.2%)
Child characteristics		
Age at enrollment (days)	145.6 + −19.8 (47 to 189)	145.3 + −18.8 (92 to 197)
Gender; female	102 (51)	103 (53.1)
Birth weight (kg)	3.2 + −0.4 (1.8 to 4.4)	3.2 + −0.5 (1.2 to 4.5)
Weight at enrollment (kg)	7.2 + −0.9 (3.8 to 10)	7.2 + −0.9 (4.9 to 9.9)
Only child	86 (43)	98 (50.5)
Feeding		
Breastfeeding	48 (24)	64 (33.2)
Formula type	118 (59)	100 (51.8)
Mixed	34 (17)	29 (15)
Parents characteristics		
Mothers’ degree		
Primary school	59 (29.5)	50 (25.9)
Secondary school	90 (45)	80 (41.5)
Degree or higher	31 (30.5)	63 (32.6)
Mother BMI (kg/m^2^)	25.8 + −4.4 (17.7 to 43.1)	25 + −4.4 (17.6 to 46.9)
Father BMI (kg/m^2^)	26.3 + −2.9 (18.7 to 34.8)	25.9 + −3.1 (17.3 to 36.7)
Mothers’ adherence to Med Diet	62.2 + −13.9 (28.6 to 92.9)	62.7 + −13.5 (28.6 to 85.7)

**Table 2 nutrients-14-02486-t002:** Differences between groups in primary and secondary end points with respect to ITT population. Variables are described by mean ± standard deviation (min to max) or absolute frequencies (percentage) ° Odds Ratio (95% Confidence Interval); § Mean difference (95% Confidence Interval).

	Control Arm (*n* = 200; 50.8%)	Med Diet Arm (*n* = 194; 49.2%)	Treatment Effect (95% CI)	*p* Value
Children				
KidMed score ≥ 8; *n* (%)	62 (34.3%)	105 (59.3%)	2.8 (1.82 to 4.3) °	<0.001
KidMed score	6.0 ± 2.9	7.5 ± 3	1.5 (0.9 to 2.1) §	<0.001
BMI (kg/m^2^)	16.4 ± 2	16.2 ± 1.5	−0.16 (−0.54 to 0.22) §	0.413
Overweight; *n* (%)	29 (17.7)	18 (11)	0.58 (0.31 to 1.1) °	0.094
Obese; *n* (%)	2 (1.2)	0 (0)	NA	NA
Parents				
Mothers’ adherence to Med Diet	61.5 ± 13.8 (28.6 to 92.2)	70.3 ± 18.2 (28.6 to 92.2)	8.6 (4.9 to 12.3) §	<0.001

**Table 3 nutrients-14-02486-t003:** Clinical ad demographical characteristics of the children analyzed for microbiota stratified by treatment arm. Variables are described by mean ± standard deviation (min to max), median [25th; 75th percentile] or absolute frequencies (percentage).

	Control Arm (*n* = 25; 49%)	Med Diet Arm (*n* = 26; 51%)
Gender; female	14 (56%)	17 (65.4%)
C-Section	9 (36%)	9 (34.6%)
Weight at T36 (kg)	14.7 ± 1.6 (12 to 18)	15.3 ± 1.9 (12.3 to 20.5)
Feeding at T36		
Breastfeeding	0 (0%)	1 (3.9%)
Formula type	4 (16.0%)	3 (11.5%)
Cow milk	21 (84.0%)	22 (84.6%)
KidMed score at T36	6.6 ± 2.1 (3 to 11)	8 ± 3 (0 to 12)
Age at microbiota evaluation	4.0 ± 0.6 (3.3 to 5.5)	4.3 ± 0.6 (3.4 to 5.6)
Number of cumulative antibiotics prescriptions at microbiota evaluation	4 (1 to 9)	5 (2 to 7)

## Data Availability

Data described in the manuscript, code book, and analytic code will be made available upon request pending application and approval of the corresponding author.

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
