# Peer review of "The Effect of Weaning with Adult Food Typical of the Mediterranean Diet on Taste Development and Eating Habits of Children: A Randomized Trial"

_nutrients, 2022, doi:10.3390/nu14122486_

Round 1
Reviewer 1 Report
Thank you for giving me the opportunity to read this interesting and original study. However, the manuscript contains many inaccuracies, and the study has certain biases that at least need to be discussed.
Major comments
- The abstract should be modified according to the comments below
- In the title and manuscript: Most of the time, weaning is for the Codex, the WHO and the EFSA the way to stop feeding with mother's milk and to start eating other food, including formula. But otherwise, it means the introduction of solid foods”. So I do think important to give which definition you used as early as possible in the introduction
- Please, explain also in the introduction what a Mediterranean diet is, since there seem to be several definitions in the literature. You may see Davis et al. Nutrients 2015.
- At the end of the introduction, you set the goal of “prevention of diseases related to SCI later in life”. But your population samples were only followed up to 3 years of age. This is very contradictory.
- It would be interesting to have the energy and nutrient intakes (at least the macronutrients) at each visit, giving the median and the IQR, as recommended by the EFSA. the MD per se cannot be considered as a factor of influence on the BMI except to consider the differences in nutritional intake.
- There is a methodologic bias. Indeed, the microbiota studies were performed at the age of 4, but the follow-up of the children and of their diet was up to 3 years. What were the differences in feeding in the interval?
- What were the antibiotics prescriptions in both groups. This also may influence the resulting microbiota
- Another bias is that no microbiota studies were performed at birth, nor at inclusion, nor at the age of 3 which is the end of the protocol design. On the other end the prevalence of C-section delivery in both groups was not given. Subsequently I don’t thing any causal relationship can be asserted.
- Also, no information is given about the type of milk given during the follow-up, and the two groups must be compared on these items (breast milk, formula, whole or half-skimmed cow’s milk?
- The discussion should involve all the biases and shortcomings detailed above.
- Line 377 and following, the difference in the range of weaning age should be discussed. In the control group the minimum is as early as 47 days, even if the ANOVA on mean age showed no difference. It should be more accurate to give the median age and the IQR. There was also significant trend in the control group to be less breast fed (p= 0.048).
- The discussion should involve the weaknesses and limitations of the studies which are numerous.
- A conclusion is needed
Minor comments
Line 53 and 54, please give the meaning of WD and MD as first cited in the manuscript
Line 68, specify years
Line 124, cooking
Line 132, usually for standard deviation the abbreviation SD is used
Line 179, specify how these cases involved in the microbiota studies were selected
Line 250-253, this sentence is confusing since you are talking about adherence to MD in both groups. Same comment for line 266.
Figure 2, please give the % of consumers in all bars
Line 317, This sentence should be modified: this is a medium-term study
Line 398 and following are somewhat speculative and not based on the study
Author Response
To Reviewer 1
We would like to thank very much the reviewer for giving us important suggestions in order to ameliorate our paper. We tried to answer to all the points he/she underlined.
It follows a point by point reply to these remarks.
- 1) In the title and manuscript: Most of the time, weaning is for the Codex, the WHO and the EFSA the way to stop feeding with mother's milk and to start eating other food, including formula. But otherwise, it means the introduction of solid foods”. So I do think important to give which definition you used as early as possible in the introduction
We thank the reviewer for this appropriate observation. We explained at the first mention of the word “weaning” what we intended with this expression both in the Abstract (line 23) and in the Introduction (lines 79-80).
Abstract:
“We aimed to verify the effects of a complementary weaning using adult foods typical of MD….” ® “We aimed to verify the effects of weaning (i.e. the introduction of solid foods in infants previously fed only with milk) using adult foods typical of MD…”
Introduction:
“The purpose of our work is to study if complementary feeding with natural foods of the MD is a simple and easily reproducible approach to increase in children (and their families) the adherence to MD…” ® “The aim of our work was to verify if weaning (i.e. the introduction of solid foods in infants previously fed only with milk, currently known as complementary feeding) with natural foods of the MD……”
- 2) Please, explain also in the introduction what a Mediterranean diet is, since there seem to be several definitions in the literature. You may see Davis et al. Nutrients 2015.
We thank the reviewer for allowing us to properly define our schema of Mediterranean Diet. In the Introduction we inserted the reference of Nutrients 2015, and better detailed it in the Material and Methods paragraph which MD we used (MD Foundation, 2011).
Moreover, at line 181-185 the following sentence was added “Due to the low age of children, some differences were applied in respect of the original MD schema: nuts only after one year of age; total exclusion of red wine, only one serving of vegetables, fruit and cereal per meal and reduced number of eggs to 1-2 weekly. Finally, our weaning schema did not include any sweet”.
- 3) At the end of the introduction, you set the goal of “prevention of diseases related to SCI later in life”. But your population samples were only followed up to 3 years of age. This is very contradictory.
We agree with the reviewer on this point and we deleted the sentence at the end of the introduction.
- 4) It would be interesting to have the energy and nutrient intakes (at least the macronutrients) at each visit, giving the median and the IQR, as recommended by the EFSA. the MD per se cannot be considered as a factor of influence on the BMI except to consider the differences in nutritional intake.
These data would definitely be interesting but, unfortunately, we were not able to get them due to the organizing difficulties during the COVID-19 pandemic.
- 5) There is a methodologic bias. Indeed, the microbiota studies were performed at the age of 4, but the follow-up of the children and of their diet was up to 3 years. What were the differences in feeding in the interval?
We thank the reviewer for allowing us to better clarify this very important point. We added in the paragraph 3.4 at lines 392-394 that the way of feeding children between 36 months of age and the time of feces analysis was based on the same recommendations used in the first three years of follow-up. Moreover we added a new table (Table 3) in which the main characteristics of children who were involved in the microbiota analysis are reported, stratified by treatment group.
- 6) What were the antibiotics prescriptions in both groups. This also may influence the resulting microbiota
This information is reported in the new Table 3.
- 7a) Another bias is that no microbiota studies were performed at birth, nor at inclusion, nor at the age of 3 which is the end of the protocol design.
The reviewer was right in underlying this limit that we now reported explicitly in the Discussion
“Although we did not analyze the gut microbiota at baseline, we observed that children weaned with a MD….” ® “Although we did not analyze the gut microbiota at baseline, nor at three years of age, we observed that children weaned with a MD…..”
- 7b) On the other end the prevalence of C-section delivery in both groups was not given. Subsequently I don’t thing any causal relationship can be asserted.
As before, this information is reported in the new Table 3.
8) Also, no information is given about the type of milk given during the follow-up, and the two groups must be compared on these items (breast milk, formula, whole or half-skimmed cow’s milk?
As before, this information is reported in the new Table 3.
- 9) The discussion should involve all the biases and shortcomings detailed above.
- 11)The discussion should involve the weaknesses and limitations of the studies which are numerous
We added in the discussion (lines 478-483) a paragraph reporting the main limitations of the study.
- 10) Line 377 and following, the difference in the range of weaning age should be discussed. In the control group the minimum is as early as 47 days, even if the ANOVA on mean age showed no difference. It should be more accurate to give the median age and the IQR. There was also significant trend in the control group to be less breast fed (p= 0.048).
We thank the reviewer for this comment; the value of 47 days in the control group refers to a single child and, apart from that, the distribution of this variable in the two groups was very similar (and symmetric). Actually, the medians (IQR) were 146 (134; 156) days in control group and 148 (133; 156) days in the experimental group. With respect to feeding, we agree that there was a slight larger prevalence of breast-feeding in the experimental group, but we do not consider this difference as “clinically” relevant for the main findings. Moreover, due to the randomized design we did not believe necessary to assess the significance of the difference between groups with respect to baseline conditions.
- 12) A conclusion is needed
A Conclusion was added at the end of the Discussion (lines 535-538)
Minor comments
Line 53 and 54, please give the meaning of WD and MD as first cited in the manuscript
Thanks. Text was modified accordingly.
Line 68, specify years
Thanks. Text was modified accordingly.
Line 124, cooking
Thanks. Text was modified accordingly.
Line 132, usually for standard deviation the abbreviation SD is used.
Thanks. Text was modified accordingly.
Line 179, specify how these cases involved in the microbiota studies were selected
Thanks. This information was added in line 393-395.
Line 250-253, this sentence is confusing since you are talking about adherence to MD in both groups. Same comment for line 266.
Thanks for this observation. Actually, our aim was to measure the adherence to Mediterranean Diet in all the children with the hypothesis that a weaning schema based on fresh fruits and vegetables even with green leaves, fish and avoiding any kind of sweet would reinforce the adherence to MD more than a traditional weaning. This is why we administered the Kidmed questionnaire to both groups.
Figure 2, please give the % of consumers in all bars.
Thanks. Figure was modified accordingly.
Line 317, This sentence should be modified: this is a medium-term study.
Thanks. Text was modified accordingly.
Line 398 and following are somewhat speculative and not based on the study
Thanks. We rephrase the whole paragraph (lines 526-534) to explicitly point out the speculative nature of this observation.
Reviewer 2 Report
The authors present an interesting paper evaluating the effects of introducing a Mediterranean style diet during the weaning period on adherence to such a diet and the resultant effects on the gut microbiota during the preschool years. However, the methods in particular required considerable clarification in order to understand specifically what was involved in this study. Specific comments as follows:
- I found the introduction a little disjointed and think it could be rw-written to focus on the study at hand, rather than trying to cover a much broader range of topics as it currently does. Does the Med diet affect the microbiota? Can children adhere to a Med diet? Has any research been undertaken in the preschool age group? I think this would give the reader a much better idea of what we do and don't know about weaning with Med diets and why you devised your specific objectives.
- Lines 91-91 - please explain further what you mean by the pediatricians defining their anthropometric measurements and Med diet schema - surely it was important for this to be consistent? The next few paragraphs would suggest there was some conformity but it is not currently clear.
- Did any stratification by feeding status (breast-fed/formula-fed) occur at baseline? If not, how might this have impacted your findings?
- Should this should be a cluster randomised trial given that children who attend a specific pediatric office are more likely to be similar than those at different practices? The sample size calculations would suggest this is not the case. Please argue.
- It is unclear how involved the intervention was. Did any mention of the med diet just occur opportunistically during any visit to the pediatrician's office (and thus vary from participant to participant)? Or were there some standard assessments? If so, when did these occur during the intervention? How long did they typically last/ What was discussed?
- Lines 366-9. I would reword this - Figure 3 does not suggest a difference in dietary adherence at 12 months of age (i.e. "weaning") but does at 4 years of age (when the microbiota were measured).
Minor comments
1. I would restrict the use of abbreviations - particularly non-common ones such as WD. This is also not defined the first time it is used (introduction).
Author Response
To Reviewer 2
We would like to thank very much the reviewer for giving us important suggestions in order to improve our paper. We tried to answer to all the points he/she underlined.
It follows a point by point reply to these remarks.
- 1) I found the introduction a little disjointed and think it could be re-written to focus on the study at hand, rather than trying to cover a much broader range of topics as it currently does. Does the Med diet affect the microbiota? Can children adhere to a Med diet? Has any research been undertaken in the preschool age group? I think this would give the reader a much better idea of what we do and don't know about weaning with Med diets and why you devised your specific objectives.
Thank for this suggestion. We rewrote the introduction focusing more on the topics of the study.
- 2) Lines 91-91 - please explain further what you mean by the pediatricians defining their anthropometric measurements and Med diet schema - surely it was important for this to be consistent? The next few paragraphs would suggest there was some conformity but it is not currently clear.
We thank the reviewer for allowing us to better clarify this point. We added a reference in which is described the method we used for measuring children and we reported that each child was always measured by the same family pediatrician thus reducing any measurement bias (lines 89-91). “All pediatricians preliminarily defined their methodology of measuring kids’ weight and length” ®“All pediatricians preliminarily defined their methodology of measuring kids’ weight and length according to standard procedures (Alan D Rogol and Gregory F Hyden, J Pediatr 2014;164:S1-S14)” and each child was always measured by the same pediatrician.
.
- 3) Did any stratification by feeding status (breast-fed/formula-fed) occur at baseline? If not, how might this have impacted your findings?
The primary end-point of our study was to verify if complementary feeding with natural foods of the MD increases in children the adherence to MD; differences in microbiota was a secondary end-point and thus we did not consider necessary to adopt such stratification. Due to the randomized design, we do not expect this should have impact on the main findings of the study.
- 4) Should this be a cluster randomised trial given that children who attend a specific pediatric office are more likely to be similar than those at different practices? The sample size calculations would suggest this is not the case. Please argue.
We thank the reviewer for this comment. Actually, as the unit of randomizations was child and not pediatrician, we did not adopt a cluster design neither in the sample size calculation nor in the statistical analysis. We agree that a cluster design would have reduced contamination bias among participants but the open label design, that was required by our trial, would have, in turn, introduced a risk of biased recruitment.
- 5) It is unclear how involved the intervention was. Did any mention of the med diet just occur opportunistically during any visit to the pediatrician's office (and thus vary from participant to participant)? Or were there some standard assessments? If so, when did these occur during the intervention? How long did they typically last/ What was discussed?
We agree with the reviewer and detailed more this point. At line 201-203, we add the phrase: “The reinforce typically last five minutes and had the purpose to verify if correct eating habits were still kept focusing on the consume of fresh fruits and vegetables even with green leaves, fish and avoiding any kind of sweet”.
- 6) Lines 366-9. I would reword this - Figure 3 does not suggest a difference in dietary adherence at 12 months of age (i.e. "weaning") but does at 4 years of age (when the microbiota were measured).
We agree with the reviewer on this point and we deleted the sentence at lines 366-69.
Minor comments
I would restrict the use of abbreviations - particularly non-common ones such as WD. This is also not defined the first time it is used (introduction).
Thanks for the suggestion. We tried to restrict as possible the use of abbreviations.
Round 2
Reviewer 2 Report
The authors have made some changes to the manuscript but in several cases, their reasoning and thought processes are still difficult to follow.
1. I don't believe the introduction has really been improved - it is still lacking flow. Many of the sentences are ok on their own - but they still do not "fit" together.
2. You won't have eliminated bias - you hope that each paediatrician is consisent within their own measures, but this doesn't mean they are consistent between one another.
3. Randomisation should indeed control for both known and unknown confounders, but stratification can be useful in situations where you have variables that you believe can influence the outcome (such as breastfeeding) that have the potential to be different across your intervention groups.
4. Your answer to 4 did not make sense to me. You have an intervention that is delivered by pediatricians. Children who attend a certain clinic are likely more similar than children who attend a different clinic. Your pediatricians might also have delivered the intervention differently - insufficient detail is provided regarding fidelity to know if this is indeed the case. To me, this should definitely be a cluster randomised trial the way it is presented, so the limitation of this should be acknowledged.
5. Ok
6. Ok
7. Please remove all abbreviations to make the paper easier to follow.
Author Response
We would like to thank the reviewer for his/her helpful comments and suggestions. In the following we tried to answer point-by-point to them.
I don't believe the introduction has really been improved - it is still lacking flow. Many of the sentences are ok on their own - but they still do not "fit" together.
We appreciate your comment. We now rewrote the introduction trying to give more coherence to the text.
You won't have eliminated bias - you hope that each paediatrician is consisent within their own measures, but this doesn't mean they are consistent between one another.
We may understand with your worries. Indeed, a centralized evaluation of children weight and length/height would have strengthened our results. However the setting of our trial, i.e. the family pediatricians, did not allow such approach.
Randomisation should indeed control for both known and unknown confounders, but stratification can be useful in situations where you have variables that you believe can influence the outcome (such as breastfeeding) that have the potential to be different across your intervention groups.
Thank for this remark. During the design of the study we may have underestimated the risk of unbalance by chance in this important "prognostic" factor. However, as a sensitivity analysis, we performed the analysis of the primary outcome (Kidmed score ≥8) using a logistic regression model adjusted by feeding status and the results were as follows: OR: 2.8, 95% CI: 1.81 to 4.33; p<0.001 which are almost equal to those obtained in the unadjusted analysis.
Your answer to 4 did not make sense to me. You have an intervention that is delivered by pediatricians. Children who attend a certain clinic are likely more similar than children who attend a different clinic. Your pediatricians might also have delivered the intervention differently - insufficient detail is provided regarding fidelity to know if this is indeed the case. To me, this should definitely be a cluster randomised trial the way it is presented, so the limitation of this should be acknowledged.
We appreciate this remark. We now acknowledged this limitation in the discussion "Another limitation is the decision to perform a randomization at children level and not at pediatrician level by adopting a cluster design that would have reduced the risk of contamination."
Please remove all abbreviations to make the paper easier to follow.
Thanks for your suggestion. We removed all the abbreviations